# Effect of Exogenous pH on Cell Growth of Breast Cancer Cells

**DOI:** 10.3390/ijms22189910

**Published:** 2021-09-14

**Authors:** Sungmun Lee, Aya Shanti

**Affiliations:** 1Healthcare Engineering Innovation Center, Department of Biomedical Engineering, Khalifa University of Science and Technology, Abu Dhabi 127788, United Arab Emirates; aya.shanti@ku.ac.ae; 2Khalifa University’s Center for Biotechnology, Khalifa University of Science and Technology, Abu Dhabi 127788, United Arab Emirates

**Keywords:** breast cancer, exogenous pH, cell cycle, apoptosis, reactive oxygen species

## Abstract

Breast cancer is the most common type of cancer in women and the most life-threatening cancer in females worldwide. One key feature of cancer cells, including breast cancer cells, is a reversed pH gradient which causes the extracellular pH of cancer cells to be more acidic than that of normal cells. Growing literature suggests that alkaline therapy could reverse the pH gradient back to normal and treat the cancer; however, evidence remains inconclusive. In this study, we investigated how different exogenous pH levels affected the growth, survival, intracellular reactive oxygen species (ROS) levels and cell cycle of triple-negative breast cancer cells from MDA-MB-231 cancer cell lines. Our results demonstrated that extreme acidic conditions (pH 6.0) and moderate to extreme basic conditions (pH 8.4 and pH 9.2) retarded cellular growth, induced cell death via necrosis and apoptosis, increased ROS levels, and shifted the cell cycle away from the G0/G1 phase. However, slightly acidic conditions (pH 6.7) increased cellular growth, decreased ROS levels, did not cause significant cell death and shifted the cell cycle from the G0/G1 phase to the G2/M phase, thereby explaining why cancer cells favored acidic conditions over neutral ones. Interestingly, our results also showed that cellular pH history did not significantly affect the subsequent growth of cells when the pH of the medium was changed. Based on these results, we suggest that controlling or maintaining an unfavorable pH (such as a slightly alkaline pH) for cancer cells in vivo could retard the growth of cancer cells or potentially treat the cancer.

## 1. Introduction

Breast cancer is the most common type of cancer in women and is the leading cause of cancer death in females worldwide [1,2]. Breast cancer most commonly originates from the inner lining of milk ducts in the breasts or from the lobules that supply the ducts with milk [3]. Like many other cancerous cells, breast cancer cells have the ability to metastasize or invade nearby tissues and spread to distant regions of the body, where they become increasingly life-threatening. Not all breast cancer cells are the same. Some express specific receptors such as estrogen receptors (ER), progesterone receptors (PR), or human epidermal growth factor receptor 2 (HER2), while others do not [4]. Approximately 70–80 percent of breast cancers are hormone receptor positive and express ER or PR [5]. Hormone-receptor-positive breast cancers can be treated with hormone therapy drugs. Breast cancer cells that do not express any of these three receptors are called triple negative and are usually highly aggressive. In addition, they cannot be treated with hormone therapy drugs [6].

Regardless of their type, breast cancer cells differ from normal cells in many ways. For instance, they express more receptors than normal cells do, such as the receptor tyrosine kinases, which mediates fundamental cellular functions including proliferation, survival, adhesion, and differentiation [7,8]. In addition, they exhibit an altered metabolism and tend to reduce the extracellular pH (pH_e_) in their environments [9].

Alterations to the extracellular and intracellular pHs in tissues affect cellular function and play an important role in cancer development [9,10]. Normal cells tightly maintain intracellular pH (pH_i_) to near-neutral values via ion transport proteins [11]. Normal differentiated adult cells have a pH_i_ of ~7.2 and a pH_e_ of ~7.4. However, cancer cells create a reversed pH gradient with their pH_e_ being more acidic than normal cells. Cancer cells have a higher pH_i_ of >7.4 and a lower pH_e_ of ~6.7–7.1 [9,10]. This reversed pH gradient is thought to be due to an altered metabolism in cancer cells. Otto Warburg et al. first reported abnormal anaerobic glycolysis and metabolic alterations in cancer cells [12]. In cancer cells, pyruvate (the end product of glycolysis) is converted into lactate (an end product of anaerobic respiration) instead of citrate, despite the presence of oxygen [13]. Lactate secretion by cancer cells, in turn, leads to extracellular acidification [14,15]. Moreover, cancer cells increase the expression and the activation of transporters and pumps such as the Na^+^/H^+^ exchanger (NHE-1), the H^+^-lactate co-transporter, and the proton pump (H^+^-ATPase), which increases the secretion of H^+^ into the extracellular environment of the cells, further contributing to extracellular acidification [16,17]. Therefore, cancer cells have a reversed pH gradient with a slightly elevated intracellular pH and a slightly lowered extracellular pH [18,19].

Low extracellular cancer pH plays a significant role in drug resistance and in the promotion of invasive growth and metastases [20,21]. In particular, low extracellular pH increases the expression of genes that encode matrix-degrading enzymes and proangiogenic factors and upregulates the activity of metastatic effectors such as serine proteases and angiogenic factors [22].

Based on the reversed pH gradient of cancer cells, some researchers suggest that alkaline treatment might be an effective complementary treatment for those who suffer from cancer [23,24]. The basic idea of alkaline treatment is that the reversed pH gradient of breast cancer cells can be restored by the systemic administration of alkaline compounds such as sodium bicarbonate (NaHCO_3_) [25]. However, there is still no strong evidence to prove that a diet of alkaline food can manipulate whole body pH or that it has a direct impact on cancer. Sodium bicarbonate treatment was shown to reduce colonization of the lymph nodes, but did not significantly affect the levels of circulating tumor cells [26]. In order to apply alkaline treatment to cancer cells more efficiently, it is crucial to understand the direct effect of different pH levels on cancer cells.

In this study, we investigated the effect of exogenous pH on the growth, death (via either apoptosis or necrosis), intracellular reactive oxygen species (ROS) levels and cell cycle of MDA-MB-231 cells, a triple-negative breast cancer cell line characterized by high aggressiveness and invasiveness.

## 2. Results

Cancer cells, including breast cancer cells, exhibit a reversed pH gradient. Cancer cells have a lower extracellular pH of ~6.7–7.1 and a higher intracellular pH of 7.4, while normal cells have a higher extracellular pH of 7.4 and a lower intracellular pH of 7.2 [9,10]. Therefore, some studies suggest that the treatment of cancer cells by alkaline compounds can effectively eradicate cancer [23]. However, there is still no strong evidence to prove that the treatment of cancer cells by alkaline compounds can actually destroy cancerous cells. In order to understand the effect of the different pH_e_ on breast cancer cells, we incubated triple-negative breast cancer cells (MDA-MB-231) in various pH media, and we monitored their growth profile, death, intracellular ROS levels, cell cycle and extracellular pH changes.

### 2.1. pH Changes of the Culture Medium with or without MDA-MB-231 Cells

We monitored pH changes of the medium during the incubation of cells in a humidified, 37 °C, 5% CO_2_ incubator. As shown in Figure 1, across the five different pHs (pH 6.0, pH 6.7, pH 7.4, pH 8.4, and pH 9.2), MDA-MB-231 cells lowered extracellular pH with time, and the environment of the cells became more acidic than the initial starting condition. Figure 1f demonstrated that the medium changed color over time, with all samples losing pink color. DMEM includes phenol red as a pH indicator. The color of phenol red is yellow below pH 6.8, and it is orange-red around pH 7.4, which is the physiological pH. Phenol red turns bright pink when the pH is over 8.2. DMEM also includes sodium bicarbonate, NaHCO_3_, as a buffer to stabilize pH and requires a 5–10% CO_2_ environment to maintain physiological pH.
NaHCO_3_ ↔ Na^+^ + HCO_3_^−^
H_2_CO_3_ ↔ H^+^ + HCO_3_^−^

NaHCO_3_ dissociates into sodium ion (Na^+^) and bicarbonate ion (HCO_3_^−^). In the bicarbonate buffering system, pH is maintained via Le Chatelier’s principle. When the pH of the system decreases, the increased H^+^ ion concentration drives the equation to the left. Similarly, a decrease in H^+^ ion concentration drives the equation to the right.

The partial pressure of carbon dioxide (pCO_2_) in arterial blood is between 35 mmHg and 45 mmHg. When mammalian cells are cultured, the CO_2_ concentration (pCO_2_) in a cell culture incubator is usually set at 5% (*v*/*v*) or 38 mmHg, that is 5% of 760 mmHg.

When CO_2_ is dissolved in water, it forms carbonic acid (H_2_CO_3_) and H_2_CO_3_ then freely dissociates into ions.
CO_2_ + H_2_O ↔ H_2_CO_3_ ↔ H^+^ + HCO_3_^−^

The buffering system in the medium can resist pH changes, and the pH of the medium should remain stable. However, pH disturbances are inevitable in a live-cell culture system due to cell metabolism [27]. When the disturbance exceeds the buffering capacity, the pH of the medium will change [27]. The fluctuation of pH usually leads to the acidification of the growth medium. As seen in Figure 1, media in five pHs—pH 6.0, pH 6.7, pH 7.4, pH 8.4, and pH 9.2—tended to be acidified in the incubator over time; however, acidification was accelerated at all pHs when MDA-MB-231 cells were present. 

### 2.2. Effect of Different pH Levels on MDA-MB-231 Cell Growth

MDA-MB-231 cells were cultured in five different pHs of Dulbecco’s Modified Eagle Medium (DMEM): pH 6.0, pH 6.7, pH 7.4, pH 8.4, and pH 9.2. Initial cell density was 2 × 10^5^ cells/well in 1 mL of medium at five different pHs. As seen in Figure 2a, MDA-MB-231 cells incubated in pH 6.7, pH 7.4, and pH 8.4 media became confluent in 5 days, albeit at slightly different growth rates, while cells incubated in pH 6.0 and pH 9.2 media did not. Although cells incubated at pH 6.0 and pH 9.2 were growing, they were not growing as fast as those in other pHs of medium (Figure 2b). Interestingly, pH 6.0 prevented MDA-MB-231 cells from attaching to the surface of a 12-well plate. All cell counts at designated time points in five different pHs are shown in Figure 2b.

In order to precisely identify the acidic pH conditions that prevented MDA-MB-231 cells from attaching to the surface of the well plates, cells were incubated in DMEM media at five different pHs: pH 6.00, pH 6.15, pH 6.30, pH 6.45, and pH 6.60. As seen in Figure 3, cells incubated at pH 6.30 and higher adhered to the surface of well plates while those incubated at a pH lower than 6.30 failed to do so.

As seen in Figure 2, MDA-MB-231 cells grew faster in pH 7.4 medium than they did in pH 9.2 medium. In order to discover if this change in growth rate could be reversed, MDA-MB-231 cells were cultured in pH 7.4 and 9.2 media then, once they were stable in each pH, their pHs were changed. Specifically, two samples in pH 7.4 and pH 9.2 media were kept at the same respective pHs, while two other samples had the pHs of their media switched from pH 7.4 to pH 9.2 or from pH 9.2 to pH 7.4. As seen in Figure 4, when cells experienced a pH change from pH 7.4 to pH 9.2, they exhibited the same cell growth profile as those experiencing a continuous pH of 9.2. When cells experienced a pH change from pH 9.2 to pH 7.4, their growth at day 1 was slow; however, after day 1, they exhibited same growth profile as those experiencing a continuous pH of 7.4. These results demonstrated that the current pH of the medium was more important for determining cell growth than initial seeding medium. The pH history did not significantly affect the cell growth. This suggests that controlling or maintaining unfavorable pH conditions for cancer cells in vivo could retard their growth or potentially treat the cancer.

### 2.3. Effect of Different pH Levels on MDA-MB-231 Cell Death

Cell death in five different pHs was measured by double staining of Annexin V and PI, and detecting the fluorescence of both Annexin V and PI using a flow cytometer. Apoptotic cells expose phosphatidylserine on the outer leaflet of the plasma membrane and Annexin V specifically binds to phosphatidylserine in apoptotic cells [28]. PI selectively penetrates the cell membranes of dead cells and has fluorescence when binding to the DNA of dead cells [29]. Therefore, Annexin V is a used for detecting early cellular apoptosis and PI is used to detect cellular necrosis or late apoptosis.

Figure 5 shows cell death at different pHs. The flow cytometry plots were divided into four regions labelled Q1, Q2, Q3 and Q4. Healthy cells were negative for both Annexin V and PI and they were labeled Q1 in the flow cytometry plot. Early apoptotic cells were positive for Annexin V only and they were labeled as Q2. Necrotic cells were positive for PI only and they were labelled as Q3. Apoptotic cells were positive for both Annexin V and PI and were labelled Q4.

MDA-MB-231 cells were regarded as healthy if they showed no sign of apoptosis or necrosis. The healthy cell population of MDA-MB-231 cells at pH 6.7 and pH 7.4 was over 90%; however, the healthy cell population at pH 6.0, pH 8.4, and pH 9.2 was 48.7%, 70.2%, and 55.7%, respectively. In an alkaline pH (pH 8.4 and pH 9.2), more cells were early apoptotic than in an acidic pH (pH 6.0). In an acidic pH (pH 6.0), more cells were late apoptotic or necrotic than in an alkaline pH (pH 8.4 and pH 9.2). The cell death process of MDA-MB-231 cells was accelerated when they were in an extremely acidic pH (pH 6.0) compared with when they were in an alkaline pH (pH 8.4 and pH 9.2).

### 2.4. Effect of Different pH Levels on Intracellular MDA-MB-231 Reactive Oxygen Species

Apoptosis or necrosis can be induced by many factors, such as ROS levels, cytochrome c, Fas, and/or the tumor necrosis factor (TNF) family [30]. These factors are involved in caspase-mediated apoptosis [31]. ROS are oxygen-based free radicals such as hydrogen peroxide (H_2_O_2_), superoxide (·O_2_^−^), hydroxyl radical (OH), and singlet oxygen (^1^O_2_). Overexpressed ROS can cause cell death by damaging DNA, RNA, and proteins [32,33]. The ROS levels of MDA-MB-231 cells in different pHs were measured by 5-(and-6)-chloromethyl-2′,7′-dichlorodihydrofluorescein diacetate, acetyl ester (CM-H_2_DCFDA) and dihydroethidium (DHE). CM-H_2_DCFDA can detect ROS including H_2_O_2_ and DHE is can detect intracellular superoxide. In Figure 6, pH 6.0 induced the highest intracellular ROS levels in MDA-MB-231 cells. Intracellular ROS levels in MDA-MB-231 cells incubated in pH 8.4 and pH 9.2 were higher than those of cells incubated in pH 6.7 and pH 7.4. MDA-MB-231 cells in acidic or basic conditions increased the ROS levels, causing more apoptosis and necrosis.

### 2.5. Effect of Different pH Levels on MDA-MB-231 Cell Cycle

Cancer is a disease of uncontrolled cell division and inappropriate cell proliferation which is associated with the cell division cycle. The cell cycle has five phases, including G0 (resting phase), G1 (gap phase 1), S (DNA synthesis), G2 (gap phase 2), and M (mitosis) [34]. Figure 7 demonstrates that most MDA-MB-231 cells were at the G0/G1 phase in pH 7.4 medium. The cell population in the S phase increased in an alkaline pH (pH 8.4 and pH 9.2) and the cell population in the G2/M phase increased in an acidic pH (pH 6.0 and pH 6.7).

## 3. Discussion

Breast cancer remains one of the leading causes of death in women worldwide [1,2]. Triple-negative breast cancer, in particular, is associated with aggressive histology, poor prognosis and unresponsiveness to usual hormone therapy, thereby leading to the short survival of its patients [35]. Like many other types of cancer, triple-negative breast cancer is shown to lower the pH of its extracellular environment, potentially to promote its invasiveness and metastasis [36]. In fact, the slightly acidic extracellular environment pH of cancer cells has been shown to foster endothelial-to-mesenchymal transition and active cell migration, induce the degradation of the extracellular matrix and promote angiogenesis [37]. In light of this, some studies have suggested the use of alkaline therapy to reverse the pH gradient and stop cancer progression [38,39]. However, there is still no conclusive data on the effectiveness of this therapy in treating cancer.

To better understand the effectiveness of alkaline therapy for cancer treatment, the effect of different extracellular pH levels on cancer cells should be thoroughly investigated. In this study, we investigated the way different exogenous pH levels affect MDA-MB-231 triple-negative breast cancer cells. We cultured MDA-MB-231 cells in five different pHs of medium—pH 6.0, pH 6.7, pH 7.4, pH 8.4, and pH 9.2—then examined, over time, their growth profile, death, intracellular ROS levels, cell cycle and extracellular pH changes.

Our results indicate that extreme extracellular acidic conditions (pH 6.0) retard the growth of MDA-MB-231 cells, increase intracellular ROS to very high levels, cause significant cell death via necrosis and apoptosis and shift the cell cycle from the G0/G1 phase to the G2/M phase. Interestingly, slightly acidic conditions (pH 6.7) increase the growth rate of MDA-MB-231 cells (compared to a physiological pH of 7.4), decrease intracellular ROS levels, do not cause any significant cell death and shift the cell cycle from the G0/G1 phase to the G2/M phase. These results, in particular, explain why cancer cells tend to reduce the extracellular pH from 7.4 to ~6.7–7.1; slightly acidic conditions give the cells the ability to maintain high survival and growth rates while shifting their cell cycle to the G2/M phase to proliferate and produce more cancer cells [9,10]. Additionally, some studies show that slightly acidic conditions promote cancer metastasis and induce drug resistance [20,21].

Our results also indicate that MDA-MB-231 do not favor alkaline conditions, as cells incubated in pH 8.4 and pH 9.2 media had a slower growth rate, higher ROS levels, increased cell death and had a slightly shifted cell cycle from the G0/G1 phase to the S phase. These results support other studies claiming that alkaline treatment could be a potential treatment of cancer, or at least a complementary one [24,26,38,39,40,41].

Our results also demonstrate a very interesting aspect of the effect of exogenous pH on MDA-MB-231, that is the minimal effect of pH history on cancer growth. When cells experienced a pH change from 7.4 to 9.2, they showed the same growth profile as those experiencing a continuous pH of 9.2. Similarly, when cells experienced a pH change from 9.2 to 7.4, they showed same growth profile as those experiencing a continuous pH of 7.4 after day 1. These results imply that the current pH of the medium is more important for determining cell growth than the initial seeding medium and that the pH history does not significantly affect the subsequent cell growth. Therefore, we suggest here that if we could control or maintain an unfavorable pH, such as slightly alkaline pH, for cancer cells in vivo, then we could retard the growth of cancer cells or potentially treat the cancer.

## 4. Materials and Methods

### 4.1. Cell Culture

MDA-MB-231 (ATCC, Manassas, VA, USA) cells were cultured in Dulbecco’s Modified Eagle Medium (DMEM) (Gibco, Fisher Scientific, Waltham, MA, USA) or Rosewell Park Memorial Institute (RPMI-1640) medium (Gibco) supplemented with 10% (*v/v*) fetal bovine serum (Gibco), 1% penicillin (100 U/mL) and streptomycin (100 µg/mL) (Gibco). Cells were incubated at 37 °C and 5% CO_2_.

### 4.2. pH Conditioned Media Preparation

To obtain media at specific pH levels (pH 6.0, pH 6.7, pH 7.4, pH 8.4, and pH 9.2), drops of 0.1 M HCl (Sigma, St. Louis, MO, USA) or 0.1 M NaOH (Sigma) were added gradually to the media until the desired pH level was reached. pH was monitored using a pH meter (ORION STAR A211, Thermo Scientific, Waltham, MA, USA).

### 4.3. Cell Proliferation Assay and pH Monitoring

MDA-MB-231 cells were detached from tissue culture flasks using trypsin (Gibco), resuspended in five different pHs (pH 6.0, pH 6.7, pH 7.4, pH 8.4, and pH 9.2) of DMEM, then seeded on 12-well plates at a density of 2 × 10^5^ cells/well in 1 mL of medium. The pHs of the different DMEM solutions were monitored by measuring the pH of the different solutions every day for 5 days using a pH meter (ORION STAR A211, Thermo Scientific, Waltham, MA, USA). The growth of MDA-MB-231 cells was also monitored by counting the cells every day for 5 days using a haemacytometer. Cells were resuspended with PBS containing 0.4% trypan blue (Gibco) for viable cell counts with a haemacytometer. Total viable cell counts were performed every day without changing media for up to 5 days.

### 4.4. Analysis of Apoptosis and Necrosis with Flow Cytometry

MDA-MB-231 cells were incubated at five different pHs (pH 6.0, pH 6.7, pH 7.4, pH 8.4, and pH 9.2) of DMEM for 5 days. Cells were then harvested and washed with 1 mL PBS. For the detection of apoptotic and necrotic MDA-MB-231 cells, cells were labeled by adding propidium iodide (PI) (2 μL of 1 mg/mL) and Annexin V-FITC to each sample. Samples were mixed gently and incubated for 15 min at room temperature in the dark. After 15 min, cells were centrifuged at 500× *g* for 5 min at 4 °C, resuspended in 100 μL PBS, then immediately analyzed with a BD Accuri C6 flow cytometer (BD Accuri™ C6 cytometer, BD Bioscience, San Jose, CA, USA). A minimum of 10,000 cells were analyzed.

### 4.5. Measurement of Hydrogen Peroxide and Superoxide

MDA-MB-231 cells were incubated at five different pHs (pH 6.0, pH 6.7, pH 7.4, pH 8.4, and pH 9.2) of DMEM for 5 days. Cells were then washed with PBS and stained by either 5 μM 5-(and-6)-chloromethyl-2′,7′-dichlorodihydrofluorescein diacetate, acetyl ester (CM-H_2_DCFDA) (Invitrogen, Carlsbad, CA, USA) for measuring H_2_O_2_ or 5 μM dihydroethidium (DHE) (Invitrogen, Carlsbad, CA, USA) for measuring superoxide. After 20 min, cells were washed 3 times with ice cold PBS. Fluorescence was measured by an Infinite^®^ 200 Pro microplate reader (Tecan Trading AG, Zurich, Switzerland) using a laser for either CM-H_2_DCFDA (λ_ex_/λ_em_ = 488/530 nm) or DHE (λ_ex_/λ_em_ = 518/605 nm).

### 4.6. Cell Cycle Analysis with Flow Cytometry

For cell cycle analysis, MDA-MB-231 cells cultured in different pHs were collected, washed 2–3 times in PBS, fixed in cold 70% ethanol for 30 min at 4 °C, washed again in PBS, stained with 50 µg/mL PI and finally analyzed by a flow cytometer (BD Accuri™ C6 cytometer, BD Bioscience) using a laser for PI (λ_ex_/λ_em_ = 488/605 nm). Cells were treated with 50 µL of a 100 µg/mL ribonuclease to ensure that only DNA, not RNA, was stained.

### 4.7. Statistics

Each bar graph represents the mean  ±  standard deviation of at least three independent experiments. Statistical analysis was performed using Student’s *t*-test, comparing each treatment to cells at a pH_e_ of 7.4 unless otherwise mentioned. In all statistical analyses, *p* < 0.05 (*) was considered significant.

## 5. Conclusions

In this study, we investigated the effect of different exogenous pH levels on the growth, survival, intracellular ROS levels and cell cycle of MDA-MB-231 triple-negative breast cancer cells. Our results demonstrated that extreme acidic conditions (pH 6.0) and moderate to extreme basic conditions (pH 8.4 and pH 9.2) retarded cellular growth, induced cell death via necrosis and apoptosis, increased ROS levels, and shifted the cell cycle away from the G0/G1 phase. However, slightly acidic conditions (pH 6.7) increased cellular growth, decreased ROS levels, did not cause significant cell death and shifted the cell cycle from the G0/G1 phase to the G2/M phase. This, in fact, seems to explain why cancer cells usually favor acidic conditions over neutral ones. Based on these results, we suggest that controlling or maintaining unfavorable exogenous pH levels, such as slightly alkaline pH, for cancer cells in vivo, could retard the growth of cancer cells or potentially treat the cancer.

## Figures and Tables

**Figure 1 ijms-22-09910-f001:**
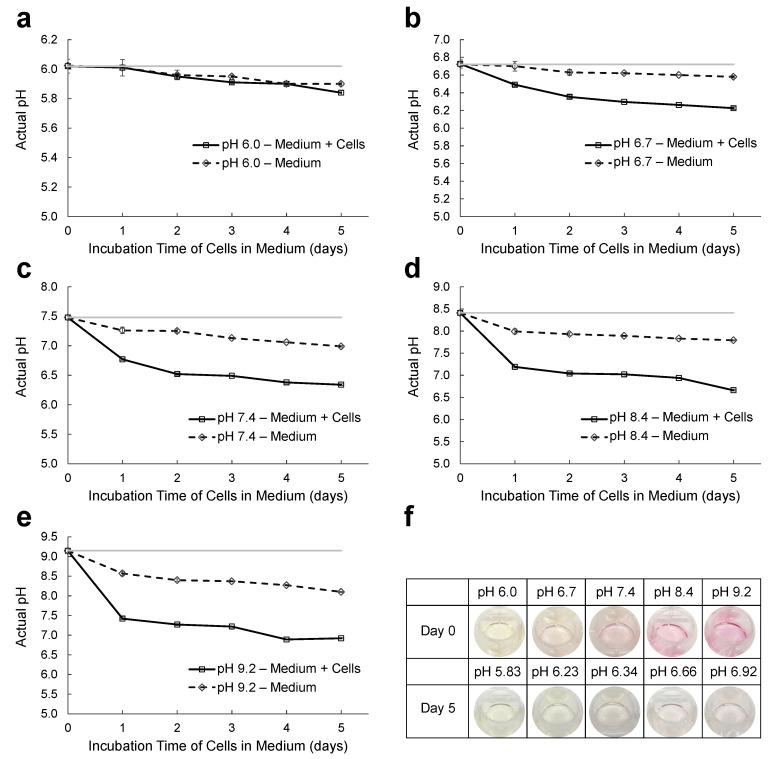
Extracellular pH profiles of MDA-MB-231 cells over time. MDA-MB-231 cells were seeded in a 12 well plate and they were incubated at five different pHs, (**a**) pH 6.0; (**b**) pH 6.7; (**c**) pH 7.4; (**d**) pH 8.4; and (**e**) pH 9.2 of DMEM. They were maintained at 37 °C in an incubator and pHs were measured every day. Medium only (dotted line), MDA-MB-231 cells in different pH medium (solid black line), and reference line (solid gray line). (**f**) Medium color of MDA-MB-231 cells in 12 well plates at day 0 and day 5.

**Figure 2 ijms-22-09910-f002:**
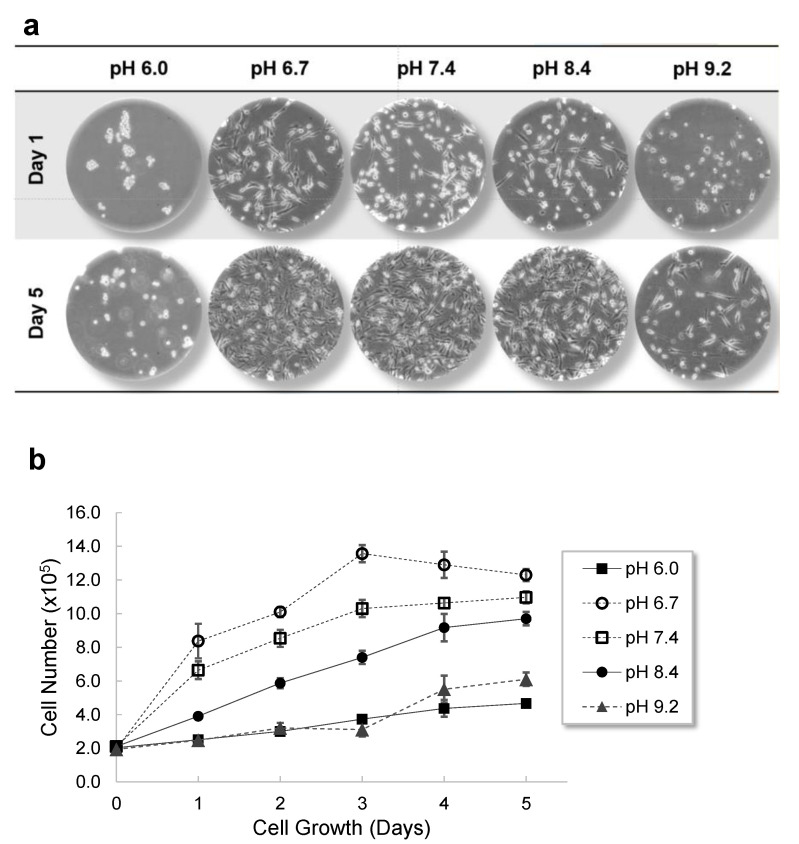
The effect of external pH on the growth of MDA-MB-231 cells, a metastatic human breast cancer cell line. MDA-MB-231 cells were seeded in a 12-well plate and they were incubated at five different pHs of DMEM: pH 6.0, pH 6.7, pH 7.4, pH 8.4, and pH 9.2. (**a**) Cells were imaged by a light microscope on day 1 and day 5. (**b**) The number of cells was also counted everyday by a hemocytometer.

**Figure 3 ijms-22-09910-f003:**
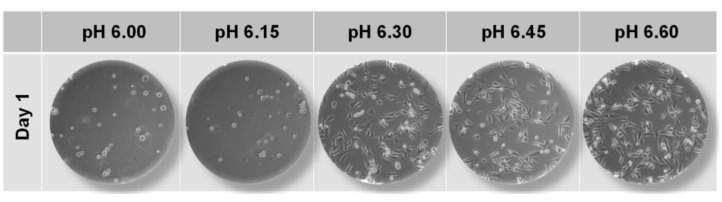
The effect of acid pHs on the growth of MDA-MB-231 cells. MDA-MB-231 cells were seeded in a 12-well plate and they were incubated at five different pHs of DMEM: pH 6.00, pH 6.15, pH 6.30, pH 6.45, and pH 6.60. Cells were imaged by an Olympus IX70 microscope using a 10× objective on day 1.

**Figure 4 ijms-22-09910-f004:**
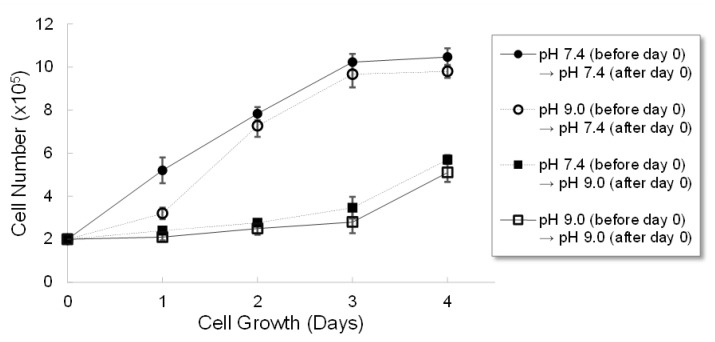
The effect of a medium change on the cell growth of MDA-MB-231 cells. MDA-MB-231 cells were grown in pH 7.4 or pH 9.0 DMEM. The cells in each pH were trypsinized and splitted in two different pHs, pH 7.4 and pH 9.0, respectively, at day 0. Total viable cell counts were performed every day without changing media for up to 4 days. pH 7.4 → pH 7.4 (black circles), pH 9.0 → pH 7.4 (white circles), pH 7.4 → pH 9.0 (black squares), and pH 9.0 → pH 9.0 (white squares).

**Figure 5 ijms-22-09910-f005:**
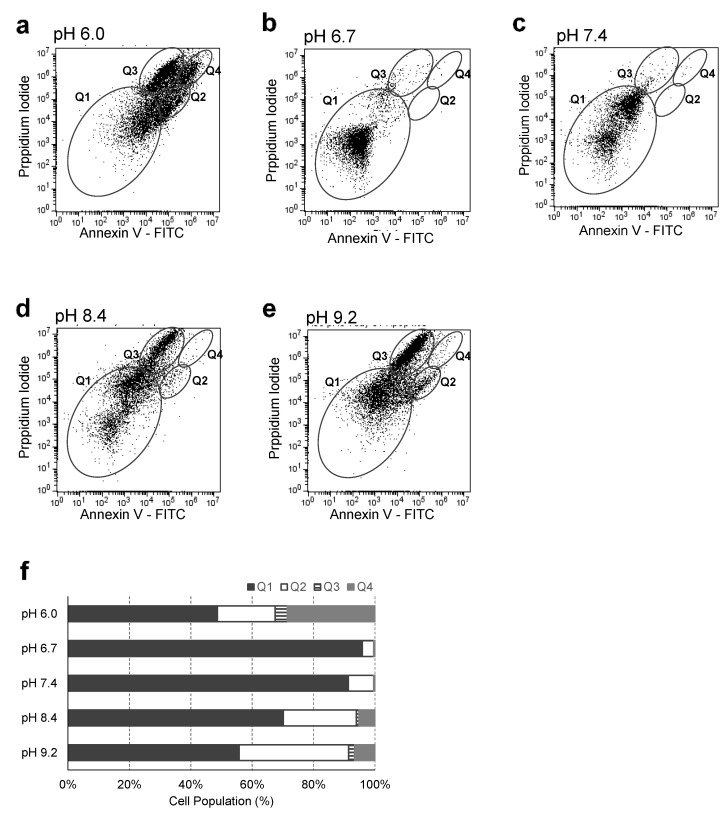
Apoptosis or necrosis analysis of MDA-MB-231 cells in different pHs. MDA-MB-231 cells were seeded in a 12-well plate and they were incubated at five different pHs of DMEM: (**a**) pH 6.0; (**b**) pH 6.7; (**c**) pH 7.4; (**d**) pH 8.4; and (**e**) pH 9.2. Cells were stained with Annexin V-FITC and PI, and apoptosis or necrosis was measured by flow cytometry assay. (**f**) Data in the dot plots were analyzed by Q1, Q2, Q3, and Q4.

**Figure 6 ijms-22-09910-f006:**
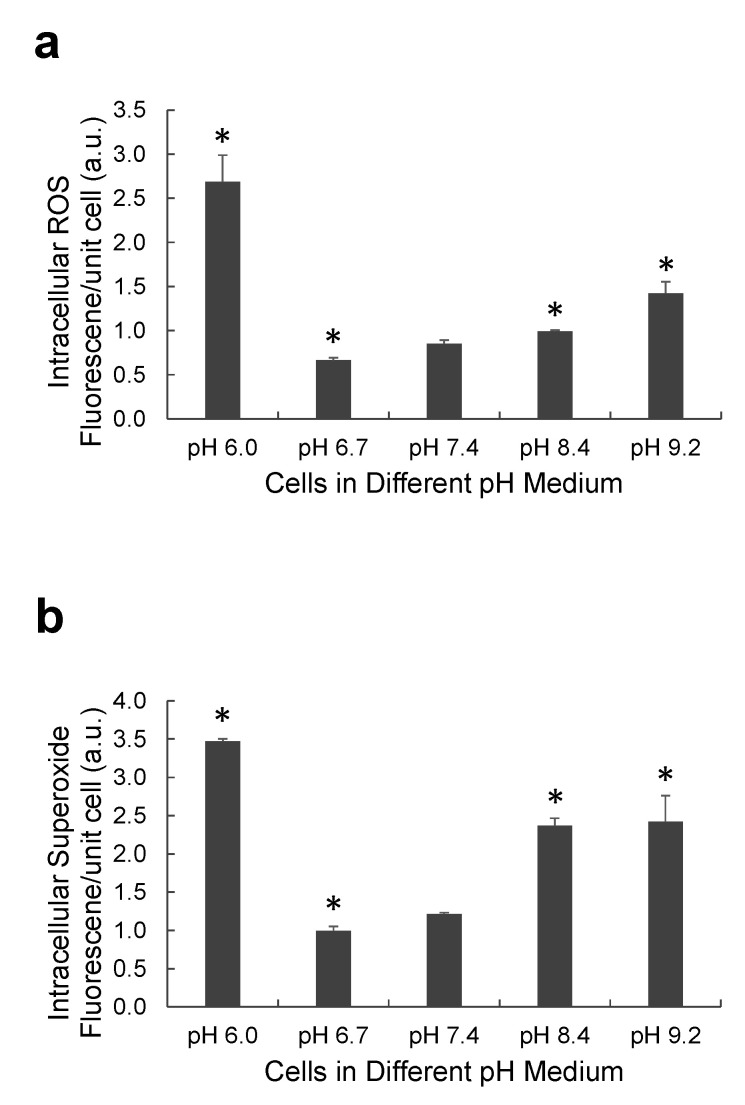
The effect of pH on the levels of intracellular reactive oxygen species (ROS) in MDA-MB-231 cells. MDA-MB-231 cells were seeded in a 12-well plate and they were incubated at five different pHs of DMEM: pH 6.0, pH 6.7, pH 7.4, pH 8.4, and pH 9.2. (**a**) Intracellular ROS or H_2_O_2_ is measured by CM-H2DCFDA (*p* < 0.05 (*) from the control pH 7.4). (**b**) The level of intracellular superoxide is also obtained by DHE (*p* < 0.05 (*) from the control pH 7.4).

**Figure 7 ijms-22-09910-f007:**
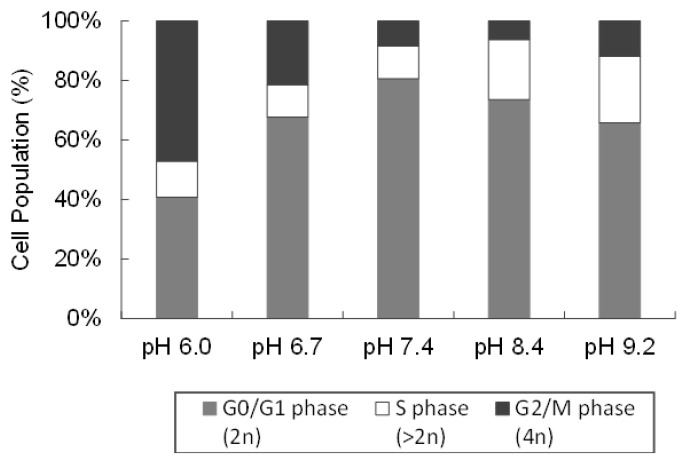
Cell cycle analysis of MDA-MB-231 cells in different pHs of medium. MDA-MB-231 cells were seeded in a 12-well plate and they were incubated at five different pHs of DMEM—pH 6.0, pH 6.7, pH 7.4, pH 8.4, and pH 9.2—for 5 days. The cell cycle (the distribution of cells in the G0/G1, S and G2/M phases) was measured by a flow cytometer after staining the DNA using propidium iodide.

## Data Availability

Data available upon request.

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
