# Peer review of "Effect of Exogenous pH on Cell Growth of Breast Cancer Cells"

_ijms, 2021, doi:10.3390/ijms22189910_

Round 1
Reviewer 1 Report
This study shows the pH response of breast cancer cell line MDA-MB-231 with conditioned pH media.
The major findings of the study is the differences of response in the pHs.
While interesting point, there are several concerns in the presented work.
Comments
1. The authors present the pH response of breast cancer cell line. It shows interesting phenotypes. The authors perform cell culture with pH conditioned media. MDA-MB-231 cell line is mainly used in the manuscript. The authors should use other cell lines in the pH response. The results of other cell lines should not be needed the same response, however; the authors should show that the pH control experiment have also work with other cells.
2. In FACS analysis for detection of apoptosis, the fractions are divided into 4 fractions. It has the potential to be sufficient to divide into 2 groups. It is needed to re-examine the results. In addition, it is needed early phase of pH response(Day1-2).
3. The authors should show the details of pH conditioned media and culture conditions. The adjustment of these media should be clearly in the materials and methods section.
4. MDA-MB-231 cells have the potential to metastasize to other organs. In the human or animal tissues have differential pH microenvironment (Zhang et al., J Phys Materials, 2019). There are many breast cancer metastatic models which shows differentially metastatic proliferation(Nakayama et al., Int J Mol Sci., 2021). For example, previous research shows differentially proliferation of breast cancer cell lines in the metastatic organ (Kuroiwa et al., Cancers, 2020). From viewpoint of pH biology, the authors should discuss the proliferative activities with the pH in metastatic organs.
5. pH 6 should be revised pH6.0 in the manuscript.
Reviewer 2 Report
Dear Authors:
In the manuscript Shanti et al., the authors show the effect of exogenous pH on cell growth of breast cancer cells, which may provide a novel potential therapeutic for breast cancer. I would give a few suggestions.
1.Some parts and paragraphs are difficult to read and should be rewritten.
1) Page 2, line 60-61:"Therefore, cancer cells have a reversed pH gradient with a 60 slightly elevated intracellular pH and a slightly lowered extracellular pH." The word "cancer","pH" should be consistent with the original format.
2) Page 2, line 67:"some research suggests that..." please correct the grammatical mistake.
3) Page 2, line 85-86:"...effectively eradicate cancer[23]."[23]" should be consistent with the original format.
4) Page 2, line 89-91:"we monitored, in time, their growth profile, their death, their intracellular ROS, their cell cycle and their extracellular pH changes." Please delete "in time" and "thier"s, just leave one "thier".
2.Some citations are missing
1) Page 7, line 208-209:"Over-expressed ROS can cause cell death by damaging DNA, RNA, and proteins." There is another review also showed that increased ROS level may cause breast cancer development by damaging DNA.(Please cite Chen et al. Semin Cancer Biol. 2020 Oct 6 doi: 10.1016/j.semcancer.2020.09.012.)
3.The manuscript needs linguistic improvement.
4.Please explain the criteria or reason you choose the gradient "pH 6.0, pH 6.7, pH 7.4, pH 8.4, and pH 9.2" for this experiment.
5.Please add a conclusion at the end of article.
Best regards
Round 2
Reviewer 1 Report
The authors addressed all my concern.
Reviewer 2 Report
Authors made correction according to my previous suggestions. Strongly recommend for publishing.